# Fludarabine, High-Dose Cytarabine and Idarubicin-Based Induction May Overcome the Negative Prognostic Impact of *FLT3*-ITD in *NPM1* Mutated AML, Irrespectively of *FLT3*-ITD Allelic Burden

**DOI:** 10.3390/cancers13010034

**Published:** 2020-12-24

**Authors:** Paola Minetto, Anna Candoni, Fabio Guolo, Marino Clavio, Maria Elena Zannier, Maurizio Miglino, Maria Vittoria Dubbini, Enrico Carminati, Anna Sicuranza, Sara Ciofini, Nicoletta Colombo, Girolamo Pugliese, Riccardo Marcolin, Adele Santoni, Filippo Ballerini, Luca Lanino, Michele Cea, Marco Gobbi, Monica Bocchia, Renato Fanin, Roberto Massimo Lemoli

**Affiliations:** 1Clinic of Hematology, Department of Internal Medicine (DiMI), University of Genoa, 16132 Genova, Italy; claviom@unige.it (M.C.); maurizio.miglino@hsanmartino.it (M.M.); ematlab@unige.it (E.C.); ematologia@unige.it (G.P.); s4582953@studenti.unige.it (R.M.); filippo.ballerini@hsanmartino.it (F.B.); s3898392@studenti.unige.it (L.L.); michele.cea@unige.it (M.C.); gobbi@unige.it (M.G.); roberto.lemoli@unige.it (R.M.L.); 2IRCCS-Ospedale Policlinico San Martino, 16132 Genova, Italy; nicoletta.colombo@hsanmartino.it; 3Division of Hematology and Bone Marrow Transplantation, Azienda Sanitaria Universitaria Integrata di Udine, 33100 Udine, Italy; anna.candoni@asufc.sanita.fvg.it (A.C.); mariaelena.zannier@asufc.sanita.fvg.it (M.E.Z.); mariavittoria.dubbini@asufc.sanita.fvg.it (M.V.D.); renato.fanin@asuiud.sanita.fvg.it (R.F.); 4Hematology Unit, University of Siena, Azienda Ospedaliera Universitaria, 53100 Siena, Italy; sicuranza4@unisi.it (A.S.); sara.ciofini@ao-siena.toscana.it (S.C.); adele.santoni@student.unisi.it (A.S.); monica.bocchia@unisi.it (M.B.)

**Keywords:** acute myeloid leukemia, *NPM1*, *FLT3*-ITD, intensified induction, minimal residual disease assessment, allogeneic stem cell transplantation

## Abstract

**Simple Summary:**

The prognostic relevance of molecular aberrations in acute myeloid leukemia (AML) has been prevalently tested in patients receiving conventional 3+7 induction. Recently, there has been a renewed interest in intensified inductions, but very few data are available on the impact of the most frequent genetic alterations with these alternative treatments. We analyzed a large multicentric cohort of younger AML patients harboring *NPM1* and *FLT3*-ITD mutations receiving an intensified fludarabine-containing regimen (FLAI). Our data suggest that in *NPM1* mut patients, FLAI may overcome the prognostic influence of co-mutated *FLT3*-ITD. The increased efficacy of this treatment seems to reduce the need for early consolidation with allogeneic transplant in double-mutated patients. Our data strongly support FLAI as an ideal backbone for combination with innovative targeted drugs, in order to further improve patients’ outcome.

**Abstract:**

The mutations of *NPM1* and *FLT3*-ITD represent the most frequent genetic aberration in acute myeloid leukemia. Indeed, the presence of an *NPM1* mutation reduces the negative prognostic impact of *FLT3*-ITD in patients treated with conventional “3+7” induction. However, little information is available on their prognostic role with intensified regimens. Here, we investigated the efficacy of a fludarabine, high-dose cytarabine and idarubicin induction (FLAI) in 149 consecutive fit AML patients (median age 52) carrying the *NPM1* and/or *FLT3*-ITD mutation, treated from 2008 to 2018. One-hundred-and-twenty-nine patients achieved CR (86.6%). After a median follow up of 68 months, 3-year overall survival was 58.6%. Multivariate analysis disclosed that both *NPM1*mut (*p* < 0.05) and ELN 2017 risk score (*p* < 0.05) were significant predictors of survival. *NPM1*-mutated patients had a favorable outcome, with no significant differences between patients with or without concomitant *FLT3*-ITD (*p* = 0.372), irrespective of *FLT3*-ITD allelic burden. Moreover, in landmark analysis, performing allogeneic transplantation (HSCT) in first CR proved to be beneficial only in ELN 2017 high-risk patients. Our data indicate that FLAI exerts a strong anti-leukemic effect in younger AML patients with *NPM1*mut and question the role of HSCT in 1st CR in *NPM1*mut patients with concomitant *FLT3*-ITD.

## 1. Introduction

An increasing number of genetic and epigenetic abnormalities have been shown to display prognostic value in acute myeloid leukemia (AML) [1,2,3]. The European Leukemia Net (ELN) implemented the risk stratification at diagnosis by integrating cytogenetics and molecular data and strongly recommended *NPM1* and *FLT3* mutational status assessment [4]. The presence of *NPM1* mutation (*NPM1* mut) reduces the negative prognostic impact of *FLT3*-ITD, which is also modulated by *FLT3*-ITD/wild-type allelic ratio [5,6]. However, most information on the prognostic impact of *NPM1* and/or *FLT3* mutations comes from trials with daunorubicin and cytarabine (“3+7”) induction [7,8,9]. The recent randomized trial by Stone et al. showed that the addition of midostaurin to conventional 3+7 induction improved the outcome of *FLT3*-ITD-positive AML patients [10]. Furthermore, the benefit of adding gemtuzumab ozogamicin (GO) to 3+7 regimen in low–intermediate-risk patients has been confirmed by a recent randomized French trial [11]. High-dose cytarabine-containing regimens (ICE, FLAI MRC, CLIA, CLAG) have been reported to achieve high complete remission (CR) rate and favorable outcome in younger AML patients, but information on their activity on specific molecular subsets is still incomplete and the role of allogeneic stem cell transplantation (HSCT) in first complete remission in this therapeutic scenario has not been defined to date [12,13,14,15,16,17,18,19]. We have already reported that, following a fludarabine-containing intensified induction, *NPM1* mutation was associated with a very high CR rate and good disease-free survival (DFS) and overall survival (OS). Moreover, the presence of *FLT3*-ITD did not negatively affect prognosis in the whole cohort of patients and HSCT in first CR did not lead to an improved outcome of non-high-risk patients [14]. However, the small size of the studied cohort did not allow to disclose which molecular subsets of patients (*NPM1* mut, *FLT3*-ITD or concomitant aberrations) may benefit the most from our intensified approach [14]. In this paper, we analyzed the impact of the two most frequent molecular aberrations in a larger cohort of AML patients, homogeneously treated with an intensified induction and consolidation therapy in three Italian hematology centers. Moreover, we evaluated the impact of HSCT in this setting

## 2. Methods

### 2.1. Study Design

This retrospective study involved, 149 patients (median age 52; range 18–65), treated with the same intensified fludarabine-containing induction between January 2008 and January 2018 in three Italian Hematology Centers, who tested positive for the *NPM1* mutation or *FLT3*-ITD mutation or both. Written informed consent for biological sample analysis and for data collection was obtained for each patient enrolled. The study was conducted according to the Declaration of Helsinki.

### 2.2. Diagnostic Workup and Molecular Analysis and Risk Assessment

Conventional cytogenetic analysis with q-banding was performed and cytogenetic abnormalities were graded according to Medical Research Council Criteria [3]. Molecular work-up was performed as per European Leukemia Net recommendation, evaluation of *FLT3*-ITD allelic burden, *TP53*, *RUNX1*, *ASXL*-1 was performed on stored samples, if not performed at diagnosis, in order to retrospectively apply European LeukemiaNet 2017 (ELN 2017) for risk definition in all patients [4]. MRD evaluation by real-time PCR for *NPM1* was performed as previously described. [5,6,20,21,22,23]. Further details on cytogenetic and molecular analysis are provided in Appendix B.

### 2.3. Treatment Schedule

Treatment included two induction courses. Induction one consisted of fludarabine 30 mg/sqm, followed 4 h later by high-dose cytarabine (2000 mg/sqm) infused in 4 h on days 1 to 5, whereas idarubicin 10 mg/sqm was added shortly after completion of cytarabine infusion on days 1, 3 and 5 (FLAI) [14]. All patients achieving hematological complete remission (CR) after FLAI received the second induction, which included high-dose cytarabine (2000 mg/sqm) on days 1 to 5, with the addition of an increased dose of idarubicin (12 mg/sqm) infused in 1 h on days 1, 3 and 5 (Ara-C + Ida) [14].

Consolidation chemotherapy included up to 3 cycles of high dose cytarabine (2000 mg/sqm in a 4-h infusion once daily on day 1 to 4, HDAC) [14].

HSCT consolidation in first CR was planned according to risk score at diagnosis, donor availability and comorbidities (see Appendix C for further details). Consolidation chemotherapy with HDAC was given until transplantation to all patients who were considered eligible for HSCT in CR1 but for any reason could not immediately proceed to transplant.

### 2.4. Response Assessment

Conventional IWG definitions were adopted for response assessment [4]. Complete Response (CR) required a blast count on bone marrow lower than 5% alongside a complete hematological recovery, defined by normal neutrophil and platelet count. Complete Response with incomplete recovery (CRi) was defined when the bone marrow criteria for CR were met but complete hematologic recovery was not achieved. Partial Response (PR) required a reduction in bone marrow blast cells higher than 50% from diagnosis, with an absolute blast count lower than 25%, without fulfilling CR or CRi criteria [4]. Bone marrow aspirate for response assessment was performed in each center as per local clinical standards. *NPM1*-based, MRD-negative CR was defined as previously described [20,21,22].

### 2.5. Statistical Analysis

Chi-Square test and Fisher’s exact were applied in order to compare dichotomous variables, whereas continuous variables were compared using Student’s *t*-test or Wilcoxon’s rank test, if normal distribution could not be confirmed. For multivariate analysis, a logistic regression model was built, including only variables with a *p* value lower than 0.100 in early univariate analysis [24].

A competing risk analysis model was built for the calculation of cumulative incidence of relapse, accounting non-relapse mortality (NRM) as a competing event. A Fine and Gray sub-distribution relative hazard method was applied for competing risk analysis, and Gray’s test was adopted for comparison. Overall Survival (OS) was calculated from the first day of induction treatment until death by any cause or until last follow-up. In order to assess the impact of transplantation in first complete remission, we built a separate landmark analysis, including only patients who were alive and still in CR at day 90. The Log-rank test was used for univariate survival analysis and all survival curves were built using the Kaplan–Meier method. Each multivariate survival analysis was performed with a Cox Proportional Hazard Model, including only variables respecting the proportional risk assumption [24]. Proportional risk assumption was checked for all variables plotting scaled Schönfeld residuals against time.

All statistical analysis, with the exception of competing risk analysis and proportional hazard assumption confirmation, were performed with IBM SPSS v22© for Linux, whereas competing risk analysis and proportional hazard assumption confirmation was performed using R statistical software (www.r-project.com) for Linux.

## 3. Results

### 3.1. Patients

One-hundred and forty-nine consecutive AML patients, with *NPM1*, *FLT3*-ITD mutation or both, treated in three Hematology Italian centers from January 2008 to January 2018, were retrospectively included in this analysis. Twenty-nine patients had isolated *FLT3-ITD* (19.5%), 59 concomitant *FLT3-ITD* and *NPM1* mut (39.6%) and 61 isolated *NPM1* mut (40.9%). ELN 2017 risk score was low in 56 (37.6%), intermediate in 51 (34.2%) and high in 42 (28.2%). Median age was 52 years (range: 18–65). All patients received the same intensified induction and consolidation. After a median of 92 days (range 84–115), 35 patients received HSCT in CR1; among them, 6, 15 and 14 were considered low, intermediate or high risk, respectively, according to ELN 2017. Patients’ characteristics are summarized in Table 1.

### 3.2. Response and Toxicities

After the first induction cycle, CR was achieved in 129/149 patients (86.6%), whereas 13/149 patients did not fulfill the CR criteria (8.7%). Sixty-day treatment-related mortality was 7/149 (4.7%), mainly due to uncontrolled bleeding (*n* = 3) or infections (*n* = 4). Overall, the vast majority of patients was able to fully receive the pre-planned dosage of induction and consolidation courses. Extra-hematological toxicity was negligible as previously reported (14).

CR rate was significantly higher in *NPM1* mut if compared to *NPM1* wt patients (90.6% and 72.4%, respectively, *p* < 0.02, Table 2). A trend towards a reduced CR rate was observed according to *FLT3*-ITD mutation (CR rate 93.4% and 81.8%, for *FLT3*-ITD-negative or -positive patients, respectively, *p* = 0.051, Table 2). Patients with low or high *FLT3*-ITD allelic burden had a similar response probability (CR rate 80.2% and 82.4% for patients with high or low allelic burden, *p* = 0.875).

Response rate was higher among isolated *NPM1* mut patients, if compared to patients with either co-mutated or isolated *FLT3*-ITD (CR rate 93.4%, 86.4% and 72.4%, respectively, *p* < 0.03, Table 2).

None of the other analyzed variables significantly impacted the CR rate.

In multivariate logistical regression analysis, *NPM1* status was the only independent predictor of response (*p* < 0.05, Table 2).

*NPM* MRD assessment was available in 63/129 CR patients (48.8%). After induction, 37/63 (58.7%) patients had *NPM* MRD-negative CR with no difference between *NPM1* mut patients with or without concomitant *FLT3* ITD (19/32, 59.4% and 18/31, 58.1%, respectively, *p* = 0.916), regardless of *FLT3* ITD allelic burden (11/18, 61.1% and 8/14, 57.1% among *NPM1* mut/*FLT3*-ITD-positive patients, with high or low *FLT3*-ITD allelic burden, respectively, *p* = 0.821).

### 3.3. Relapse and Cumulative Incidence of Relapse

After a median follow-up of 68 months (CI 95%: 55.87–80.13 months), 32 patients relapsed (24.8%).

Relapse probability was higher among patients without *NPM1* mutation (*p* < 0.01) and among high-risk patients according to ELN 2017 (*p* < 0.03). Multi-variate analysis confirmed that *NPM1* mutational status was the only predictor of relapse probability (*p* < 0.05). Relapse probability analysis is detailed in Table 3.

In competing risk analysis, 3-years cumulative incidence of relapse (CIR) was 23.6% (Appendix A).

CIR was not significantly different among *NPM1* mutated with or without concomitant *FLT3*-ITD (3-year CIR 23.8% and 19.1%, respectively, *p* = 0.698), irrespectively of allelic burden (data not shown), whereas patients with isolated *FLT3*-ITD had a significantly higher CIR (3-year CIR 42.7%, *p* < 0.05).

### 3.4. Overall Survival

In the whole cohort, 63/149 (42.3%) patients died, and 3-year OS was 58.6% (median not reached, Figure 1A).

In univariate analysis, high leukocyte count at diagnosis (*p* < 0.05) the absence of *NPM1* mutation (*p* < 0.003, Figure 1C), presence of *FLT3*-ITD (*p* < 0.01, Figure 1D) and high risk according to ELN 2017 (*p* < 0.0001, Appendix A) were correlated with significantly worse survival. Concerning *FLT3*-ITD and *NPM1* reciprocal mutational status, the presence of *FLT3*-ITD did not significantly affect survival among *NPM1* mut patients (3-year OS 52.7 and 73.4%, for *NPM1-*mutated patients with or without concomitant *FLT3*-ITD, *p* = 0.372, Figure 1B). This observation was more evident among patients aged 55 or less, where the outcome of *NPM1* mut patients, with or without concomitant *FLT3*-ITD, was almost completely superimposable (*p* = 0.924, Figure 2). The implementation of allelic burden assessment did not significantly modify those findings: 3-year OS was 71.8% and 74.1% in *NPM1*-mutated patients with or without concomitant low-burden *FLT3*-ITD (*p* = 0.758), which was not significantly different from what was observed among *NPM1*-mutated patients with concomitant high-burden *FLT3*-ITD (3-year OS 61.4%, *p* = 0.187).

*FLT3*-ITD isolated patients had a significantly worse prognosis (*p* < 0.05). Multivariate analysis disclosed that both *NPM1* mutational status and ELN 2017 risk score were significant predictors of survival (*p* < 0.05 and *p* < 0.05, respectively). Detailed OS analysis is provided in Table 4.

Landmark analysis showed that in the whole cohort patients undergoing or not HSCT in first CR did not show significantly different survival (3-year OS 58.7% and 68.9%, median 81 months and not reached, respectively, *p* = 0.348, Figure 3A).

Subgroup sub-analysis showed that performing HSCT in first CR did not result in better survival in patients with *NPM1* mutations and in patients with *FLT3*-ITD (*p* = 0.625 and 0.970, respectively, Figure 3B–C). Conversely, HSCT was beneficial for ELN 2017 high-risk patients (*p* < 0.05, Figure 3D). Further details on landmark analysis are provided in Table 4.

## 4. Discussion

In our multi-centric, real-life study including younger AML patients homogeneously treated with FLAI regimen, *NPM1* mut patients had a very good long-term outcome and concomitant *FLT3*-ITD mutation did not impact on survival, regardless of allelic burden. The outcome of *NPM1* mut AML patients seems to be at least comparable with, if not better than, that reported with conventional 3+7 regimen [7,8,9] with or without the addition of GO [11]. One possible biological explanation for the high activity of FLAI in this setting may be the higher chemo sensitivity conferred by the *NPM1* mutation. Indeed, it has been demonstrated that the cytoplasmic delocalization of *NPM1*, determined by the *NPM1* mutation, induces the reduction in the anti-apoptotic activity of *NPM1* protein and increased genomic instability [25,26]. The increased NPM1-related chemo-sensitivity and the higher intracellular cytarabine concentration following fludarabine administration may overcome the survival advantage conferred to blast cells by *FLT3*-ITD mutation [9,25,26,27,28]. This biological explanation is supported by the observation that the rate of *NPM*-MRD negative CR was not affected by concomitant *FLT3-*ITD, regardless of allelic burden.

Moreover, the crucial role of increased chemo-sensitivity, related to *NPM1* mutation, may be further sustained by the better outcome achieved in patients <55 years, where dose intensity and timing of treatment are more likely to be respected. In this age group, the survival of patients belonging to low- and intermediate ELN risk groups was, in fact, superimposable.

The retrospective nature of our study prevents us from drawing any firm conclusion from the analysis of this subset of patients and limits any comparison with prospective, randomized trials. However, some interesting results deserve to be discussed. In the recent midostaurin phase III trial, the survival advantage due to the addition of midostaurin to chemotherapy was not statistically significant when patients were censored at transplantation, thus suggesting an important therapeutic role for HSCT in first CR [10].

In a Spanish trial reporting the outcome of patients receiving intermediate-dose cytarabine-containing regimens, Pratcorona et al. showed that HSCT in first CR was not beneficial in term of relapse risk and survival for *NPM1* mutated with concomitant low-burden *FLT3*-ITD. An advantage for early transplantation was, however, evident among high-burden *FLT3*-ITD, regardless of *NPM1* status [29].

Our study confirms the good outcome achieved without frontline HSCT in the favorable group of *NPM1* mut/low-burden *FLT3*-ITD patients. With the limitation of a retrospective study, our results suggest that the FLAI regimen may reduce the need for early HSCT consolidation in the whole group of non-high-risk patients, which includes *NPM1* mut/high-burden *FLT3*-ITD patients. In this view, MRD assessment may help in identifying non-high-risk patients with suboptimal response to first induction who may benefit from early HSCT [20,22]. Furthermore, in *NPM*-mut patients, the highly sensitive PCR-based MRD evaluation is able to identify patients still in hematologic CR but with molecular relapse, thus allowing preemptive strategies of salvage therapy and subsequent HSCT consolidation [21,30]. Additionally, for patients harboring *FLT3* mutations, the recently approved, highly selective, second-generation *FLT3* inhibitors may represent an optimal bridge to transplant approach for relapsing patients [31,32,33].

Conversely, our data confirm that HSCT in first CR is the best option for ELN 2017 high-risk patients, i.e., patients with isolated *FLT3-*ITD with high allelic burden or patients with other unfavorable molecular or cytogenetic alterations [4].

## 5. Conclusions

In conclusion, with the limitations of a retrospective study, FLAI-5 seems to be an effective therapy for *NPM1* mut AML patients, regardless of *FLT3-ITD* status and may not require the application of HSCT in first CR, especially in patients achieving a rapid MRD clearance [22,29]. In AML patients with *FLT3*-ITD without *NPM1* mutation, the addition of drugs targeting *FLT3* [11,31,32,33], BCL2 [34] may be indicated. In this regard, given the very high CR rate and the good tolerability, FLAI may represent the optimal backbone for testing novel agents [15,35]. In this view, GIMEMA AML1718 trial (Eudract code 2018-000392-33) is currently evaluating FLAI plus venetoclax as induction regimen in intermediate/high-risk patients, including patients bearing *FLT3* mutations.

## Figures and Tables

**Figure 1 cancers-13-00034-f001:**
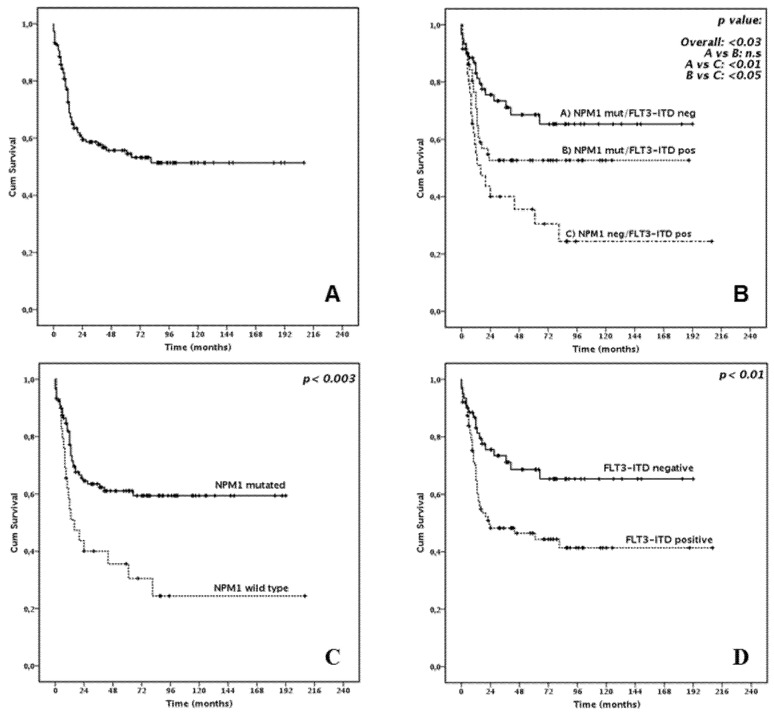
Overall Survival: (**A**) In the whole cohort (**B**) According to *NPM1*/*FLT3*-ITD status (**C**) According to *NPM1* status (**D**) According to *FLT3*-ITD status.

**Figure 2 cancers-13-00034-f002:**
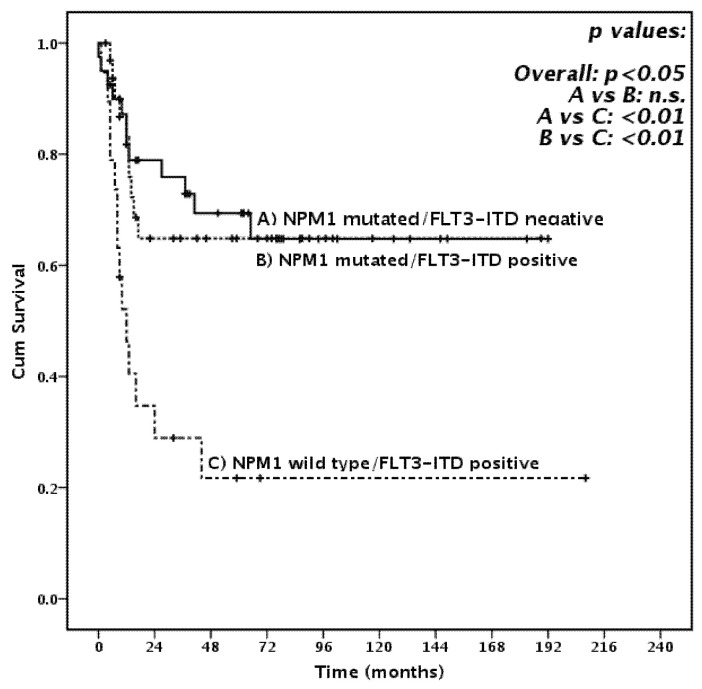
Overall Survival in patients aged 55 or less according to *NPM1* and *FLT3*-ITD status.

**Figure 3 cancers-13-00034-f003:**
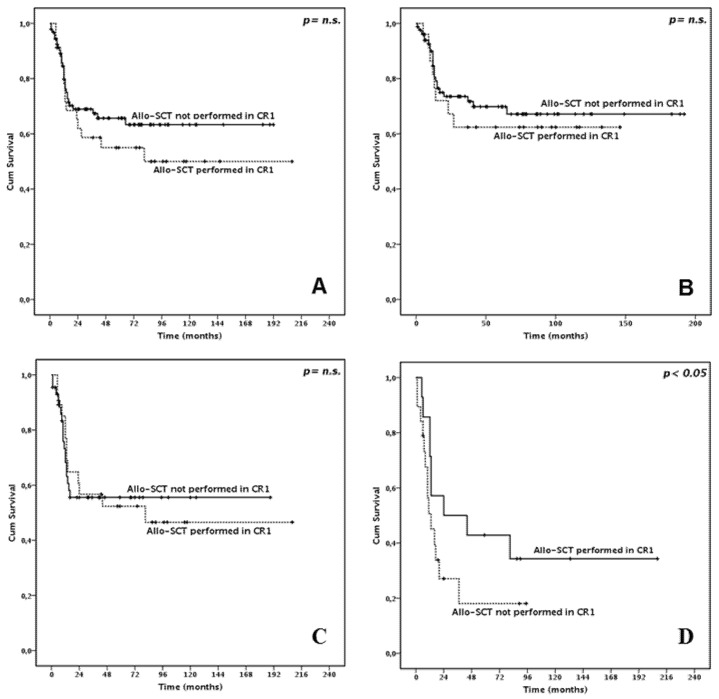
Overall Survival according to performing or not allogeneic stem cell transplantation (SCT) in first complete remission (CR1) (Landmark analysis). (**A**) In the whole cohort (**B**) In *NPM1* mutated patients (**C**) In *FLT3*-ITD positive patients (**D**) In European LeukemiaNet 2017 high risk patients.

**Table 1 cancers-13-00034-t001:** Patients’ features.

Patients’ Features		Num. (%)
**OVERALL**		149 (100%)
Age	<45 years	52 (34.9%)
	>45 years	97 (65.1%)
Sex	Male	84 (56.4%)
	Female	65 (43.6%)
Leukocytes	<30,000/μL	64 (43%)
	>30,000/μL	85 (57%)
*NPM1*	Mutated	120 (80.5%)
	Unmutated	29 (19.5%)
*FLT3*-ITD	Negative	61 (40.9%)
	Positive	88 (59.1%)
Karyotype	Intermediate	133 (89.3%)
	Unfavorable	16 (10.7%)
ELN ^1^ 2017	Low Risk	56 (37.6%)
	Intermediate Risk	51 (34.2%)
	High Risk	42 (28.2%)
*NPM1*/*FLT3*-ITD	*NPM1* mut/*FLT3*-ITD neg	61 (40.9%)
	*NPM1* mut/*FLT3*-ITD pos	59 (39.6%)
	*NPM1* wt/*FLT3*-ITD pos	29 (19.5%)

^1^ ELN = European Leukemia Net.

**Table 2 cancers-13-00034-t002:** Complete Response (CR) probability.

Patients’ Features		Num.	CR (%)	*p* (univ.)	*p* (multiv.)
**OVERALL **		149	129 (86.6)	-	-
Age	<45 years	52	47 (90.4)	0.318	-
	>45 years	97	82 (84.5)		
Sex	Male	84	73 (86.9)	1.000	-
	Female	65	56 (86.2)		
Leukocytes	<30,000/μL	64	58 (90.6)	0.209	-
	>30,000/μL	85	71 (83.5)		
*NPM1*	Mutated	120	108 (90.0)	0.019	0.012
	Unmutated	29	21 (72.4)		
*FLT3*-ITD	Negative	61	57 (93.4)	0.051	0.255
	Positive	88	72 (81.8)		
Karyotype	Intermediate	133	115 (86.5)	1.000	-
	Unfavorable	16	14 (84.5)		
ELN ^1^ 2017	Low Risk	56	53 (94.6)	0.059	0.545
	Intermediate Risk	51	43 (84.3)		
	High Risk	42	33 (78.6)		
*NPM1*/*FLT3*-ITD	*NPM1* mut/*FLT3*-ITD neg	61	57 (93.4)	0.024	-
	*NPM1* mut/*FLT3*-ITD pos	59	51 (86.4)		
	*NPM1* wt/*FLT3*-ITD pos	29	21 (72.4)		

^1^ ELN = European Leukemia Net.

**Table 3 cancers-13-00034-t003:** Relapse probability.

Patients’ Features		Num.	Relapse (%)	*p* (univ.)	*p* (multiv.)
**OVERALL **		129	32 (24.8)	-	-
Age	<45 years	47	9 (19.1)	1.000	-
	>45 years	82	23 (28.0)		
Sex	Male	73	22 (30.1)	0.109	-
	Female	56	10 (17.9)		
Leukocytes	<30,000/μL	58	14 (24.1)	0.296	-
	>30,000/μL	71	18 (25.4)		
*NPM1*	Mutated	108	23 (21.3)	0.009	0.03
	Unmutated	21	9 (42.9)		
*FLT3*-ITD	Negative	57	10 (17.5)	0.104	-
	Positive	72	22 (30.6)		
Karyotype	Intermediate	115	27 (23.5)	0.317	-
	Unfavorable	14	5 (35.7)		
ELN ^1^ 2017	Low Risk	53	7 (13.2)	0.01	0.494
	Intermediate Risk	43	9 (20.9)		
	High Risk	33	16 (48.5)		
*NPM1*/*FLT3*-ITD	*NPM1* mut/*FLT3*-ITD neg	57	10 (17.5)	0.071	-
	*NPM1* mut/*FLT3*-ITD pos	51	13 (25.5)		
	*NPM1* wt/*FLT3*-ITD pos	21	9 (42.9)		

^1^ ELN = European Leukemia Net.

**Table 4 cancers-13-00034-t004:** Overall survival analysis and landmark analysis.

Patients’ Features		Dead (%)	3-Year OS (%)	Median OS (%)	*p* (univ.)	*p* (multiv.)
**OVERALL **		63 (42.3)	58.6	NR	-	-
Age	<45 years	17 (32.7)	65.4	NR	0.086	-
	>45 years	46 (47.4)	55.2	65		
Sex	Male	41 (48.8)	53.8	44	0.108	-
	Female	22 (33.8)	65.0	NR		
Leukocytes	<30,000/μL	22 (34.4)	73.2	NR	0.034	0.569
	>30,000/μL	44 (48.2)	47.5	23		
*NPM1*	Mutated	43 (35.8)	63.5	NR	0.002	0.034
	Unmutated	20 (69.0)	40.0	16		
*FLT3*-ITD	Negative	18 (29.5)	73.4	NR	0.006	0.178
	Positive	45 (51.1)	48.2	23		
Karyotype	Intermediate	54 (40.6)	59.1	NR	0.550	-
	Unfavorable	9 (56.2)	55.6	44		
ELN ^1^ 2017	Low Risk	14 (25.0)	77.0	NR	0.000	0.048
	Intermediate Risk	19 (37.3)	58.5	NR		
	High Risk	30 (71.4)	35.2	13		
*NPM1*/*FLT3*-ITD	*NPM1* mut/*FLT3*-ITD neg	18 (29.5)	73.4	NR	0.002	-
	*NPM1* mut/*FLT3*-ITD pos	25 (42.4)	52.7	NR		
	*NPM1* wt/*FLT3*-ITD pos	20 (69.0)	40.0	16		
**LANDMARK SURVIVAL ANALYSIS**		45/129 (34.9)	63.6	NR		
All patients	HSCT in first CR	15/35 (42.9)	58.7	81	0.348	-
	No HSCT in first CR	30/94 (32.9)	68.9	NR		
*NPM1* mutated	HSCT in first CR	8/25 (32)	62.5	NR	0.625	-
	No HSCT in first CR	23/83 (37.7)	73.5	NR		
*FLT3*-ITD	HSCT in first CR	13/28 (46.4)	56.7	81	0.970	-
	No HSCT in first CR	18/44 (40.9)	55.6	NR		
ELN ^1^ 2017High Risk	HSCT in first CR	9/14 (64.3)	50	24	0.044	-
	No HSCT in first CR	14/19 (73.7)	27.1	13		

^1^ ELN = European Leukemia Net.

## Data Availability

Data available on request due to restrictions eg privacy or ethical.

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
