# Peer review of "Fludarabine, High-Dose Cytarabine and Idarubicin-Based Induction May Overcome the Negative Prognostic Impact of FLT3-ITD in NPM1 Mutated AML, Irrespectively of FLT3-ITD Allelic Burden"

_cancers, 2020, doi:10.3390/cancers13010034_

Round 1

Reviewer 1 Report

The authors propose a study about the success of therapy in a particular subgroup of AML patients. Results are interesting and well written. The conclusion is appropriate and the study certainly represents a good starting point for improving the therapeutic path of an extremely heterogeneous pathology more and more.

Minor points:

Moderate English changes are required. In particular:

-Abstract, line 73, "where" has to be corrected in "were":

-the word "addiction" has to be corrected in "addition" in several points of the manuscript.

-at the end of the "conclusions" paragraph there is one more point ".."

I recommend double-checking the spaces and punctuation throughout the text.

Author Response

We wish to thank Reviewer #1 for the positive comments and the observations. We double checked the manuscript to correct typos.

Reviewer 2 Report

This study by Minetto et al is an interesting retrospective analysis of a cohort of NPM1/FLT3 mutated AML showing the relatively good outcomes of these patients when treated with FLAI especially for patients who achieve CR with negative MRD. 

Overall the main limitation of this study is that it is a retrospective analysis which makes some of the conclusions in the discussion and throughout the manuscript really difficult to support. Thus, I feel that some of these statements should be changed to be consistent with the limitations of a retrospective study. 

Particularly:

  1. I do not think the authors can even compare the survival outcomes of this study with the 7+3+Midostaurin study as there is no comparison of the patients' population re age, comorbidities etc
  2. Mainly, I would not agree with the comment that the HSCT does not provide survival benefit in first CR for patients (especially FLT3-ITD pts) as it is impossible to make this conclusion from a retrospective analysis
  3. There is ongoing literature supporting that the FLT3 MRD may be an important factor that will play a role in the decision for transplanting patients with FLT3-ITD. Do the authors have any relevant data? If not they should include this point.
  4. I would include a paragraph with the limitations of the study
  5. I would like to see more data on the toxicities here as FLAI is a relatively toxic regimen and toxicities is an important consideration (perhaps an additional table?)

Author Response

We wish to thank Reviewer #2 for the positive comments and the insightful observations. We, of course, agree that our conclusions are limited by the retrospective nature of the study. We have therefore softened the strength of some observations and conclusion in the discussion paragraph.

1) I do not think the authors can even compare the survival outcomes of this study with the 7+3+Midostaurin study as there is no comparison of the patients' population re age, comorbidities etc

2) Mainly, I would not agree with the comment that the HSCT does not provide survival benefit in first CR for patients (especially FLT3-ITD pts) as it is impossible to make this conclusion from a retrospective analysis

4) I would include a paragraph with the limitations of the study

The discussion section was modified as follow (new sentences are highlited in bold):

"The retrospective nature of our study prevent us from drawing any firm conclusion from the analysis of this subset of patients and limits any comparison with prospective, randomized trials. However, some interesting results deserve to be discussed. In the recent midostaurin phase III trial the survival advantage due to the addition of midostaurin to chemotherapy was not statistically significant when patients were censored at transplantation, thus suggesting an important therapeutic role for HSCT in first CR [10].
In a Spanish trial reporting the outcome of patients receiving intermediate dose cytarabine containing regimens, Pratcorona et al showed that HSCT in first CR was not beneficial in term of relapse risk and survival for NPM1 mutated with concomitant low burden FLT3-ITD. An advantage for early transplantation was, however, evident among high burden FLT3-ITD, regardless of NPM1 status [29].
Our study confirms the good outcome achieved without frontline HSCT in the favorable group of NPM1 mut/low burden FLT3-ITD patients. With the limitation of a retrospective study, our results suggest that FLAI regimen may reduce the need of early HSCT consolidation in the whole group of non-high-risk patients, which includes NPM1 mut/high burden FLT3-ITD patients. Landmark analysis did not disclose a better survival for patients receiving HSCT in first CR, neither in the whole group, nor in FLT3-ITD patients".

3) There is ongoing literature supporting that the FLT3 MRD may be an important factor that will play a role in the decision for transplanting patients with FLT3-ITD. Do the authors have any relevant data? If not they should include this point.

Although FLT3-ITD mutation is one of the most common and relevant mutations in AML and the main driver of AML relapse, there is still no consensus on FLT3-ITD-based MRD evaluation in clinical practice. Existing assays for MRD in FLT3-ITD AML have not been particularly useful because of limited sensitivity. Novel MRD assay are being developed but must be standardized and validated in large cohort of patients before their incorporation in clinical practice. In our study FLT3-ITDs were detected by standard PCR as stated in the methods section, therefore our data were, unfortunately, not suitable for MRD analysis. Text was not modified.

5) I would like to see more data on the toxicities here as FLAI is a relatively toxic regimen and toxicities is an important consideration (perhaps an additional table?)

In order not to provide overlapping information with previously published data, we did not incorporated any toxicity data since we had already published safety data of FLAI regimen. (Guolo F, Minetto P, Clavio M, et al. Am J Hematol 2016, 91, 755–762). We therefore chose not to include toxicities as a main part of the study and report only sixty-days treatment-related mortality. However, in order to fulfill the Reviewer’s suggestion, we added one sentence in the results section to underline the feasibility of FLAI:

"After the first induction cycle, CR was achieved in 129/149 patients (86.6%), whereas 13/149 patients did not fulfill CR criteria (8.7%). Sixty-days treatment-related mortality was 7/149 (4.7%), mainly due to uncontrolled bleeding (n.=3) or infections (n.=4). Overall, the vast majority of patients was able to fully receive the pre-planned dosage of induction and consolidation courses. Extra-hematological toxicity was negligible as previously reported (14)".

Reviewer 3 Report

  1. What was the median and range of time to transplantation?  If the time to transplantation was longer than 90 days for many patients, then the landmark analyses based on day 90 (line 160) may still be biased.  Landmark analyses are supposed to use data as of the landmark date - so only patients who received transplant by day 90 should be included in the landmark "transplant" cohort.  It does not sound like those are what analyses were done.  
  2. Since landmark analyses can subject to reduced power from the smaller sample size, it may be nice to complement landmark analyses with time-dependent Cox model regression analyses.  
  3. Given that the 3-level combination NPM1-FLT3 variable was significant (or nearly so in table 3), why was the 3-level variable not analyzed instead of the separate NPM1/FLT3 variable?  Or why was the interaction of NPM1/FLT3 not evaluated in the multivariable models?  It is difficult to interpret the multivariable models without an interaction evaluated. 
  4. The absolute 3-year difference in OS between NPM1mut/FLT3-neg and NPM1mut/FLT3-pos is fairly large, nearly 20% (52.7% versus 73.4%).  The N in each subgroup is small and to is hard to tell whether the lack of significance is related to sample size/power or because there really is no clinically meaningful difference in the two groups.  I think given the small N and fairly large absolute difference in 3-year OS, the conclusions of the paper could be tempered some.  Especially since the 3-level variable was no evaluated in multivariable models.  
  5. I think given that retrospective nature of this analyses and the fact no data on 7+3 or 7+3+GO were presented with similarly patients, the comparisons to FLAI being "comparable, if not better" (and related phrasing in the manuscript) are too strong.  It isn't clear that any differences between these data and previously published data with other regimens might not be related to patient selection.  

Minor comments:

  1. How was this an intent-to-treat analysis (line 111).  Since there was no randomization and it is retrospective, intent-to-treat can make interpretation more challenging.
  2. Line 123 mentioned MRD for NPM1, but I wasn't clear where such data were used in the analyses.  

Author Response

What was the median and range of time to transplantation? If the time to transplantation was longer than 90 days for many patients, then the landmark analyses based on day 90 (line 160) may still be biased. Landmark analyses are supposed to use data as of the landmark date - so only patients who received transplant by day 90 should be included in the landmark "transplant" cohort. It does not sound like those are what analyses were done.
Since landmark analyses can subject to reduced power from the smaller sample size, it may be nice to complement landmark analyses with time-dependent Cox model regression analyses.

We wish to thank Reviewer #3 for the insightful comments.
All statistical analysis were carried out according to Delgado et al (reference n 24).
Median time to transplantation was 92 days (range 84-115), therefore we chose to adopt day 90 as a landmark time-point. The time to transplantation in first CR was added in the text:

"Median age was 52 years (range: 18-65). All patients received the same intensified induction and consolidation. After a median of 92 days (range 84-115), 35 patients received HSCT in CR1, among them, 6, 15 and 14 were considered at low, intermediate or high risk according to ELN 2017, respectively. Patients characteristics are summarized in Table 1".

Since the allocation to HSCT in first CR was pre-planned as soon as the risk assessment work-up was completed and the patient entered CR, and since the time range for HSCT in first CR was not significantly variable, we decided to adopt a landmark analysis (Anderson JR, Cain KC, Gelber RD. Analysis of survival by tumor response. J Clin Oncol. 1983; 1(11):710-9), whereas, probably, the most precise analysis in this context would have been performed adopting the Mantle-Byar method, which on the other hand is quite difficult and time consuming (Mantel N, Byar DP. evaluation of response-time data involving transient states: an illustration using heart-transplant data. J Am Stat Assoc. 1974; 69(345):81-6).
Anyway, we performed a time-dependent Cox proportional hazard regression model, which lead to superimposable results.

Given that the 3-level combination NPM1-FLT3 variable was significant (or nearly so in table 3), why was the 3-level variable not analyzed instead of the separate NPM1/FLT3 variable? Or why was the interaction of NPM1/FLT3 not evaluated in the multivariable models? It is difficult to interpret the multivariable models without an interaction evaluated.

As stated in methods section, only variable respecting the proportional hazard assumption were included in the Cox regression model (Cox DR. Regression models and life-tables. J R Stat Soc B. 1972; 34(2):187-220).
We acknowledge that we did not clearly disclose how this assumption was checked. Proportional hazard assumption was checked graphically with SPSS with log minus log method for all variables.
We have now double checked all variables plotting scaled Schönfeld residuals against time using the cox.zph function in R statistical software, without finding significant differences.
The method section has been modified in order to add the new analysis adopted in order to confirm proportional hazard assumption:

"Each multivariate survival analysis was performed with a Cox Proportional Hazard Model, including only variables respecting the proportional risk assumption [24]. Proportional risk assumption was checked for all variables plotting scaled Schönfeld residuals against time.
All statistical analysis, with the exception of competing risk analysis and proportional hazard assumption confirmation, were performed with IBM SPSS v22© for Linux, whereas competing risk analysis and proportional hazard assumption confirmation was performed using R statistical software (www.r-project.com) for Linux".

The absolute 3-year difference in OS between NPM1mut/FLT3-neg and NPM1mut/FLT3-pos is fairly large, nearly 20% (52.7% versus 73.4%). The N in each subgroup is small and to is hard to tell whether the lack of significance is related to sample size/power or because there really is no clinically meaningful difference in the two groups. I think given the small N and fairly large absolute difference in 3-year OS, the conclusions of the paper could be tempered some. Especially since the 3-level variable was no evaluated in multivariable models.

The 3-level variable was not evaluated in the multivariable models because it did not respect the proportional hazard assumption. We have modified the conclusions in order to reduce the emphasis:

"In conclusion, with the limitations of a retrospective study, FLAI-5 seems to be an effective therapy for NPM1 mut AML patients, regardless of FLT3-ITD status and may not require the application of HSCT in first CR, especially in patients achieving a rapid MRD clearance[29,22]. In AML patients with FLT3-ITD without NPM1 mutation the addition of drugs targeting FLT3[11,31-33], BCL2[35] may be indicated".

I think given that retrospective nature of this analyses and the fact no data on 7+3 or 7+3+GO were presented with similarly patients, the comparisons to FLAI being "comparable, if not better" (and related phrasing in the manuscript) are too strong. It isn't clear that any differences between these data and previously published data with other regimens might not be related to patient selection.

We have modified the discussion section, also in accordance with suggestion from Reviewer #2:

"The retrospective nature of our study prevent us from drawing any firm conclusion from analysis of this subset of patients and limits any comparison with prospective, randomized trials. However, some interesting results deserve to be discussed. In the recent midostaurin phase III trial the survival advantage due to the addition of midostaurin to chemotherapy was not statistically significant when patients were censored at transplantation, thus suggesting an important therapeutic role for HSCT in first CR [10].
In a Spanish trial reporting the outcome of patients receiving intermediate dose cytarabine containing regimens, Pratcorona et al showed that HSCT in first CR was not beneficial in term of relapse risk and survival for NPM1 mutated with concomitant low burden FLT3-ITD. An advantage for early transplantation was, however, evident among high burden FLT3-ITD, regardless of NPM1 status [29].
Our study confirms the good outcome achieved without frontline HSCT in the favorable group of NPM1 mut/low burden FLT3-ITD patients. With the limitation of a retrospective study, our results suggest that FLAI regimen may reduce the need of early HSCT consolidation in the whole group of non-high-risk patients, which includes NPM1 mut/high burden FLT3-ITD patients. Landmark analysis did not disclose a better survival for patients receiving HSCT in first CR, neither in the whole group, nor in FLT3-ITD patients".

How was this an intent-to-treat analysis (line 111). Since there was no randomization and it is retrospective, intent-to-treat can make interpretation more challenging.

In order to provide real-life data, we decided to include in this study all patients who were considered fit for FLAI induction, regardless if the induction treatment was actually completed (two of the patients died because of bleeding in the first 60 days actually died before FLAI was even completed). However, we do recognize that the definition can be misleading, so we removed intention to treat analysis:

"This retrospective study involved, in an intention to treat analysis, 149 patients (median age 52; range 18-65), treated with the same intensified fludarabine-containing induction between January 2008 and January 2018 in 3 Italian Hematology Centers, who resulted positive for NPM1 mutation or FLT3-ITD mutation or both".

Line 123 mentioned MRD for NPM1, but I wasn't clear where such data were used in the analyses.

NPM1 MRD data are shown in this paper in the response section (lines 183 to 187):

"NPM MRD assessment was available in 63/129 CR patients (48.8%). After induction, 37/63 (58.7%) patients had NPM MRD negative CR with no difference between NPM1 mut patients with or without concomitant FLT3 ITD (19/32, 59.4% and 18/31, 58,1%,respectively, p=0.916), regardless of FLT3 ITD allelic burden (11/18, 61.1% and 8/14, 57.1% among NPM1 mut/FLT3-ITD positive patients, with high or low FLT3-ITD allelic burden, respectively, p=0.821)".

Round 2

Reviewer 2 Report

The authors have replied to my comments.

Author Response

We wish to thank the Rewiever for helping us to improve our manuscript.

Reviewer 3 Report

The clarifications were very helpful.  Thank you.

As one last question, since you chose not to fit the 3-level variable, do you look at models with interaction terms between NPM1 and FLT3?  Given the significance of the 3-level variable, I am concerned that the models including only NPN1 and FLT3 as marginal effects cannot fully model the data patterns.  If all the interactions terms were not significant, I think this would make me much more confident in the current multivariable models.  

Author Response

We wish to thank the Reviewer for the insightful comments and for helping us to improve our manuscript.

We have double checked the interaction terms and we are able to confirm that are not significant.

Round 3

Reviewer 3 Report

Thank you for double-checking.